# Surgical Interventions in Patients Hospitalised with COVID-19. A Review of Seven Months of Experience Working in a COVID-19 Dedicated Centre

**DOI:** 10.3390/jcm10030395

**Published:** 2021-01-21

**Authors:** Justyna Rymarowicz, Michał Pędziwiatr, Piotr Major, Bryan Donohue, Karol Ciszek, Michał Nowakowski

**Affiliations:** 12nd Department of General Surgery, Jagiellonian University Medical College, 30-688 Krakow, Poland; jrymarowicz@su.krakow.pl (J.R.); piotr.major@uj.edu.pl (P.M.); karol.ciszek2@gmail.com (K.C.); mnowakowski@uj.edu.pl (M.N.); 2Department of Anaesthesia, Peterborough City Hospital, Peterborough PE3 9GZ, UK; donohue.med@gmail.com

**Keywords:** COVID-19 pandemic, surgical consultations, telemedicine, surgical care, surgical interventions, SARS-CoV-2

## Abstract

The Coronavirus Disease 2019 (COVID-19) pandemic has made changes to the traditional way of performing surgical consultations. The aim of the present study was to assess the need for surgical care performed by various surgical specialties among patients infected with COVID-19 hospitalized in a COVID-19 dedicated hospital. All surgical consultations performed for patients infected with COVID-19 in a COVID dedicated hospital in a seven month period were evaluated. Data on demographics, surgical specialty, consult reason, procedure performed, and whether it was a standard face to face or teleconsultation were gathered. Out of 2359 COVID-19 patients admitted to the hospital in the seven month period, 229 (9.7%) required surgical care. Out of those 108 consultations that did not lead to surgery, 71% were managed by telemedicine. A total of 36 patients were operated on while suffering from COVID-19. Out of them, only three patients admitted primarily for COVID-19 pneumonia underwent emergency surgery. The overall mortality among those operated on was 16.7%. Conclusions: Patients hospitalised with COVID-19 may require surgical care from various surgical specialties, especially during peaks of the pandemic. However, they rarely require a surgical procedure and only occasionally require major surgery. A significant portion of potentially surgical problems could be managed by teleconsultations.

## 1. Introduction

Coronavirus Disease 2019 (COVID-19) is an infectious disease caused by a novel betacoronavirus severe acute respiratory syndrome (SARS)-CoV-2 [1]. Its clinical manifestations resemble, to some extent, those of severe acute respiratory syndrome (SARS)-CoV-1, and include a range of symptoms like fever, cough, myalgia, diarrhoea, or headache [2]. Although the majority of infected patients present mild or no symptoms, a portion progress to respiratory failure with a high fatality rate [2]. The novel coronavirus outbreak was first reported in Wuhan, Hubei province, China in December 2019 [3]. Since then, it promptly spread to other countries. As we have been observing in the recent months, the number of people requiring hospital admission due to coronavirus infection can rise drastically over a period of several days. Between 23 and 29 November 2020, the number of new cases in Poland increased by 100,911 individuals [4]. At this time, there were on average 7.2 new cases per 10,000 habitants per day in the Malopolska voivodeship, making it the second largest epicentre in Poland. For this reason, the health care systems had to transform their facilities in order to adjust to the changing public health needs [5,6,7]. Moreover, in many countries, including Poland, because of the lack of a sufficient number of beds in the existing health care facilities, temporary coronavirus hospitals had to be created to accommodate the greatly increased number of hospitalized patients at the peak of the pandemic.

Undoubtedly, physicians such as general medicine doctors and general practitioners play the primary role in the care for COVID-19 patients. However, in the large population of hospitalized patients, there should also be a need for surgical care.

The aim of the present study was to assess the need for surgical care performed by various surgical specialties among patients infected with COVID-19 hospitalized in a COVID-19 dedicated hospital, as well as to assess the possibility of implementing teleconsultations in managing surgical consultations.

## 2. Materials and Methods

This study was conducted in the University Hospital, Cracow, Poland between 1 April 2020 and 30 October 2020. On 12 March 2020, the hospital was first transformed into a COVID-19 dedicated centre, and then from 15 September 2020 into a tertiary COVID-19 centre serving as a hospital intended to treat complicated COVID-19 cases.

All patients admitted to the hospital were tested for COVID-19 infection with a RT-PCR test.

The inpatient General Surgery; Neurosurgical; Ear, Nose, and Throat (ENT); Vascular surgery; Trauma and Orthopaedic (T&O); Maxillofacial; Gynaecology and Urology consult databases were reviewed, retrospectively searching for all patients diagnosed with COVID-19 infections who had documented surgical consultations. The theatre operating list was searched independently for patients diagnosed with COVID-19 who were operated on during the study period. In addition, patients in-hospital and outpatient clinic records were reviewed to obtain information about the complication and mortality rates. Patients who were consulted more than one time for the same condition were classified as one consultation and patients who were consulted by more than one surgical specialty for different reasons were counted as separate consultations. The consultation was classified as standard face-to face consultation if this method of consultation was performed at least once for the specific patient.

Only adult patients (above 18 years old) who were admitted to hospital and suffered from COVID-19 infection at the time of consultation were included. Patients diagnosed with COVID-19 after the consultation occurred, those considered convalescents, and those consulted in the emergency department (ED) and subsequently discharged were not included in the study.

The study was approved by the Bioethics Committee of the Jagiellonian University (1072.6120.228.2020).

The information gathered included the following:

Baseline information
Date of surgical consultation;Patient’s age;Gender;Date of the consultation.Reason for hospital admission:Related to COVID-19 infection;Due to surgical condition with COVID-19 diagnosed on screening.Type of surgical consultation:General Surgery; Neurosurgical; Ear, Nose and Throat (ENT); Vascular surgery; Trauma and Orthopaedic (T&O); Maxillofacial; Gynaecology and Urology.Reason for consultation/surgical problemMeans of consultationStandard face-to-face consultation;Teleconsultation.

For patients who underwent surgical interventions, the following additional data were gathered:

There were two distinguished procedure grades based on Bupa schedule of procedures (Bupa schedule of procedures) [8]:minor (minor or intermediate);major (major or complex major).American Society of Anaesthesiologist (ASA) grade [9];Urgency of surgery (elective, emergency);Post-operative complications;30-day mortality.

### 2.1. Statistical Analyses

The study was designed according to STROBE guidelines for observational studies.

A descriptive analysis was performed. Categorical variables were described using frequency rate and percentage. Continuous variables were described with mean and range. There were no missing data.

### 2.2. Patients Flow in the Health Care System

During the study period, patients with suspected or confirmed COVID-19 admitted to our centre were either brought into the emergency department by ambulance or transferred from other hospitals in the Malopolska voivodeship after being diagnosed with COVID-19. The hospital wards were converted into COVID-19 dedicated units with several different profiles. This included dialysis, respiratory, cardiology, surgical wards, and so on. Patients were assigned to different units depending on their need and comorbidities. The general surgical ward was transformed into a COVID-19 dedicated unit, where patients diagnosed with COVID-19 and having concomitant surgical conditions were hospitalised. The patients were managed by the multidisciplinary team comprising internal medicine doctors and surgical teams of various surgical specialties, depending on the patients’ need.

### 2.3. Management of Surgical Consultations

All consultation requests were first reviewed by the consultant surgeons and the decision was made of whether the patient required a face-to-face consultation or if an over the phone consultation could be performed. If the patient required a face-to-face consultation, it was performed by a senior clinician to minimise the number of people required to perform the consultation. Before seeing the patient, the surgeon donned FFP2/3 mask, visors, disposable scrubs, double glows, surgical gown, head cup, and theatre clogs. After the consultation, the surgeon doffed and showered in the sluice located outside the COVID ward.

Over the phone consultations were based upon diagnostic imaging, available laboratory results, imaging of the ulcer/wound uploaded into patients records if required, and information provided by the leading clinician.

## 3. Results

During the seven-month period, 2359 patients infected with COVID-19 were admitted to our hospital. Of them, 229 (9.7%) required surgical attention, with a mean age 64.7 (range 21–97), and 114 (49.8%) were female. In total, there were 77 (33.6%) general surgery, 32 (14%) urology, 27 (11.8%) neurosurgery, 25 (10.9%) T&O, 24 (10.4%) vascular surgery, 20 (8.7%) gynaecology, 17 (7.4%) ENT, and 7 (3%) maxillo-facial surgery consultations. Of those, 36 (15.7%) underwent major surgery and another 45 (19.7%) patients underwent minor surgical procedures.

From the 36 patients, 35 patients with SARS-CoV-2 infection were operated on in the emergency setting, while one patient had elective surgery (orchidectomy for testicular cancer) and, in four patients, the surgery was postponed until they had recovered from COVID-19. Thirty-three of them (91.7%) were admitted primarily because of surgical conditions and only three patients (8.3%) were admitted primarily because of COVID-19 related pneumonia and developed acute surgical symptoms during their hospital stay. The 30-day mortality rate among patients who underwent surgery was 16.7% (three craniectomy, two embolectomy, and one laparotomy patient); the ASA scores in those patients were either IV or V. Four patients (12.1%) admitted primarily with acute surgical symptoms developed COVID-19 pneumonia during the postoperative period. Details about the type of surgery and procedures performed are presented in Table 1.

A total of 141 (6%) patients required surgical consultations while they were admitted primarily owing to COVID-19 infection with no prior surgical symptoms on admission. Three patients (0.1%) required emergency surgery. One patient underwent craniectomy for subdural haematoma, which occurred during their stay in intensive care. One patient underwent embolectomy for upper limb artery thrombosis and one patient with a history of Crohn’s disease underwent laparotomy for bowel obstruction. The first two patients died within 30 days; all three patients were hospitalized primarily owing to COVID-related pneumonia. Thirty (1.3%) patients underwent minor procedures, with the most common being wound care treatment (6 patients), interventions for urinary tract obstruction (5 patients) endoscopy due to intestinal tract bleeding (4 patients), and anterior nose tamponade for epistaxis (2 patients). The remaining 108 (4.6%) patients requiring consultations did not result in a surgical procedure and 71% (77 cases) of those could be performed via telemedicine only. Further, 100% of neurosurgery, 100% of T&O, 100% of maxillofacial, 81.8% of vascular surgery, 78.5% of urology, 42.8% of gynaecology, and 39% of general surgery consultations could be successfully managed without face-to-face contact. The most common reasons for standard face-to-face consultation and via telemedicine are presented in Table 2.

Eighty-eight (3.7%) patients were hospitalized primarily owing to surgical condition with concomitant SARS-CoV-2 infection diagnosed on routine screening on admission. Thirty-three (37.5%) of them required major surgery, 15 (17%) underwent minor procedures, and 40 (45.5%) were managed conservatively.

## 4. Discussion

In response to the COVID pandemic, several aspects of medical as well as surgical care had to be transformed. Care for patients infected with COVID-19 poses additional challenges to the traditional way of performing surgical consultations. In addition, owing to infection control policies, patients diagnosed with COVID-19 are commonly admitted into separate isolation wards or into new temporary facilities where surgical assistance may not be easily accessible.

To the best of our knowledge, this is the first study exploring the surgical input of various surgical specialties in the care of patients diagnosed with COVID-19 hospitalised in the COVID dedicated centre. The appropriate management of patients suffering with COVID-19 is still under debate. The pandemic is ongoing with no signs of abating, requiring systemic solutions. The results of our study show that patients hospitalized as a result of COVID-19 infection required surgical attention. However, over three quarters of those consultations did not result in surgical intervention and, in the majority of cases, they could be performed via telemedicine only. In the seven-month period, out of over two thousand patients admitted primarily owing to COVID-19 infection, only three required emergency surgery. There were 30 minor interventions performed and the majority of them required either general surgical (wound care, endoscopy, pneumothorax drainage), ENT (epistaxis), or urology (procedures for urinary tract obstructions) intervention.

Patients who underwent surgery were predominantly admitted owing to surgical problems and SARS-CoV-2 infection was detected during screening on admission. General surgery was the most commonly involved surgical speciality, followed by neurosurgery, urology, and T&O. The relatively high number of neurosurgical interventions can be explained by the fact that our hospital is a major trauma centre for the region and a tertiary neurosurgical centre; therefore, most of the COVID-19 patients with neurosurgical problems were referred to our hospital for ongoing surgical care. Similarly, some of the patients requiring vascular surgery interventions were transferred to our hospital from other units.

At the outbreak of the coronavirus pandemic, various national and international surgical bodies published guidelines with recommendations regarding the surgical care of patients with coronavirus infection [10,11]. Moreover, Polish guidelines were released recommending that only emergency operations should be performed in COVID-19 patients and, whenever possible, non-surgical management should be implemented in order to minimise the risk of cross-infection among theatre staff [5,6]. In addition, it has been recognized that COVID-19 infection is an independent risk factor for post-operative pulmonary complications and increased mortality rate [12]. In keeping with that, all but one patient in our study were operated on in an emergency setting owing to their acute presentation. One case of elective surgery in the patient with testicular cancer could be explained by the relatively low risk of postoperative complications related to SARS-CoV-2 infection in a young asymptomatic male and the need for expedited surgery because of the possible negative outcome of postponing treatment [13,14,15]. The most common emergency procedures performed in COVID-19 infected patients were the following: laparotomy due to intestinal obstruction, embolectomy, neck of femur fracture repair, and either burr hole or craniotomy for intracranial bleeding, all of which are common surgical emergencies. Of note, the 30-day mortality rate of the patients operated on in our study was 16.7%. By contrast, the 30-day mortality rate of COVID-19 patients undergoing surgery in the large cohort study performed by COVIDSurg collaborative was 23.8% [12]. The lower mortality rate in our study could be explained by significant discrepancies in the sample sizes as well as the heterogeneities of the groups. In addition, the role of comorbidities and medication interactions were not included in our study [16,17].

There are several papers exploring the increased implementation of teleconsultation during the coronavirus pandemic in both the outpatient clinic as well as in hospital settings. They have been shown to be safe and effective in selected cases in various surgical specialties [18,19]. In our study, over a half of all consultations that did not require surgical intervention could be successfully managed by telemedicine. Furthermore, neurosurgical, T&O, and maxillofacial consultations could be solely performed via telemedicine, whereas general surgical and gynaecology consultations were mostly performed as standard face-to-face consultations, which emphasises the importance of those surgical specialities to be readily available to assistance in care of patients hospitalised owing to COVID-19 infection. Teleconsultations undoubtedly have some advantages, especially in COVID patients. For example, performing the consultation in a timely manner by avoiding spending additional time on donning and doffing, minimising the risk of SARS-CoV-2 infection among hospital staff, and using PPE resources wisely. It could be argued that performing teleconsultation instead of face-to-face assessments may compromise patients’ care and appropriate diagnosis. Therefore, the decision to complete a consultation without assessing the patients should be reserved for selected cases only.

There are several limitations to this study. Firstly, until 15 September, our hospital served as a COVID-19 centre and the vast majority of infected patients from the region were admitted solely to our centre. Whereas after that date, because of the rampant increase in coronavirus infections in the country, other hospitals also created COVID-19 wards. Only complicated cases, including those requiring surgery, were transferred to our centre, which may falsely raise the proportion of COVID-19 patients requiring surgical intervention at the end of the scrutinized period. Secondly, some consultation requests were inadequately placed, which could to some degree have increased the rate of teleconsultations. Thirdly, this is a retrospective study, so no interference could be made in the designed clinical pathways. Further, it included different surgical specialties that were at times difficult to compare. Lastly, the role of comorbidities, medication interactions, and the influence of specific medications on the clinical outcome of COVID-19 were not investigated in this study.

## 5. Conclusions

Patients hospitalised with COVID-19 infection require surgical care of various surgical specialties, especially during the peak of the pandemic. However, they rarely require a surgical procedure, and only occasionally major surgery. Furthermore, teleconsultations present a viable alternative to face-to-face consultations, especially in surgical specialties such as neurosurgery, T&O, and maxillofacial surgery. As the likelihood of eradicating COVID-19 completely is low, there will have to be designated areas for coronavirus patients in hospitals and the management of surgical care for those patients will need to be adequately addressed.

## Figures and Tables

**Table 1 jcm-10-00395-t001:** Types of surgical procedures performed on patients infected with Coronavirus Disease 2019 (COVID-19). No missing data. ENT, ear, nose, and throat; T&O, trauma and orthopaedic.

Surgical Specialty C6:E50	Type of Procedure	*n* of Procedures
Vascular surgery		
Major surgery:	Embolectomy	4
	Amputation	2
	Stent-graft implantation	1
Minor intervention:	Hemostatic wound dressing for SFA hematoma	1
	Vascular access for dialysis	1
General surgery		
Major surgery:	Laparotomy due to obstruction	5
	Laparotomy due to pelvic abscess	1
	Laparotomy due to biliary obstruction	1
	Cholecystectomy	1
	Appendectomy	3
	ERCP	3
Minor intervention:	Pneumothorax drainage	4
	Wound care	5
	Endoscopy	8
	Peritoneal dialysis access removal	1
Gynaecology		
Major surgery:	Diagnostic laparoscopy	1
Minor intervention:	Dilation and curettage for miscarriage	6
Neurosurgery		
Major surgery:	Burr hole for subdural haematoma	4
	Craniectomy for epidural haematoma	2
	Embolisation for SAH	1
T&O		
Major surgery:	Neck of femur fracture repair	5
	ORIF for ankle fracture	1
Minor intervention:	Tendon rupture repair	1
Urology		
Major surgery	Orchidectomy for testicular carcinoma	1
Minor intervention:	Nephrostomy	3
	Cystostomy	1
	Double J insertion	1
	Suprapubic catheter insertion	1
	Pyelography/cystoscopy	2
	Catheterisation with Tiemman catheter	2
	Dorsal slit of the foreskin	1
	Cystoscopy	1
Maxillofacial surgery		
Minor intervention:	Tooth extraction	2
	Wound care	1
ENT surgery		
Minor intervention:	Anterior nasal packing for epistaxis	2
	Tracheostomy tube insertion	1

SFA, superficial femoral artery; ERCP, endoscopic retrograde choleangiopancreatography; SAH, Subarahnoid hemorrhage; ORIF, Open reduction internal fixation.

**Table 2 jcm-10-00395-t002:** Most common reasons for teleconsultation and standard face-to-face consultation by surgical specialty. There were no missing data.

	**Teleconsultations**	
Surgical specialty	Reason for consultation	*n*
Vascular surgery		
	Deep vein thrombosis	6
	Superficial haematoma	3
General Surgery	Postoperative plan	1
	Preparation for elective surgery	1
	Acute cholangitis	1
	Abdominal pain	1
	Superficial haematoma	3
	Biliary colic/cholecystitis	2
	Wound care	1
	Colon cancer treatment	1
	Raised intraabdominal pressure	1
	Breast ulcer	1
	Surgical emphysema	1
Maxillofacial Surgery	Facial trauma	3
Gynaecology	Vaginal bleeding	2
	Postdelivery plan	2
	Treatment plan for PID	
ENT	Impaired hearing	2
	Sinusitis	4
	Oral candidiasis	1
	Oropharyngeal cancer treatment	2
	Upper airway track injury	2
Neurosurgery	Intracranial tumour	6
	Intracranial bleeding	7
	Spinal fracture	2
	Other	2
T&O	Neck of femur fracture	3
	Spinal fracture	3
	Shoulder fracture	5
	Knee pain	2
	Other	4
Urology	Haematuria	4
	Urinary tract obstruction	3
	Urinary tract tumour	1
	Others	5
	**Face-to-Face Consultation**	
Vascular surgery	Acute limb ischaemia	4
	AAA	1
	Other	2
General surgery	LGI bleeding	2
	Wound inspection	5
	r/o Perforation or obstruction	5
	Acute cholecystitis	1
	Other	5
Gynaecology	PID	1
	Vaginal bleeding	2
	Pregnancy control	4
ENT	Facial nerve impairment	1
	Epistaxis	1
	Facial oedema	1
Urology	Haematuria	6
	Epididymitis	1

PID, pelvic inflammatory disease; AAA, abdominal aortic aneurysm; LGI, Lower gastro intestinal tract.

## Data Availability

The data presented in this study are available on request from the corresponding author. The data are not publicly available due to privacy restrictions.

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
