# Peer review of "Surgical Interventions in Patients Hospitalised with COVID-19. A Review of Seven Months of Experience Working in a COVID-19 Dedicated Centre"

_jcm, 2021, doi:10.3390/jcm10030395_

Round 1
Reviewer 1 Report
This is an interesting article presenting data on surgical procedures in COVID-19 patients in a single center in Poland.
This is an interesting article presenting data on surgical procedures in COVID-19 patients in a single center in Poland.
- The paper needs editing. I suggest to have it revised by a native English speaker
- Table 2 is missing (only table 1 and 3 are presented in the paper)
- References to relevant articles are missing. No reference is provided in the methods section. At least references for Bupa schedule of procedures and ASA grade should be provided.
- Please revise numbers in the results section. Authors state that 36 consultations resulted in major surgery and 45 in minor surgery. However, in table 1 only 70 procedures are described (34 major and 36 minor). Similarly, numbers provided in the second paragraph of the results section are not in line with those presented in the first paragraph
Author Response
Dear Reviewer,
We appreciate the time and effort that you have dedicated to providing your valuable feedback on our manuscript. We are grateful for your insightful comments on our paper. We have been able to incorporate changes to reflect most of the suggestions provided by you. We have highlighted the changes within the manuscript body (green font).
This is an interesting article presenting data on surgical procedures in COVID-19 patients in a single center in Poland.
- Comment: The paper needs editing. I suggest to have it revised by a native English speaker.
Response:Thank you, It has been revised by a native English speaker.
- Comment: Table 2 is missing (only table 1 and 3 are presented in the paper)
Response:Thank you for noticing that, it is corrected now.
- Comment: References to relevant articles are missing. No reference is provided in the methods section. At least references for Bupa schedule of procedures and ASA grade should be provided.
Response:Thank you, this has been added accordingly.
- Comment: Please revise numbers in the results section. Authors state that 36 consultations resulted in major surgery and 45 in minor surgery. However, in table 1 only 70 procedures are described (34 major and 36 minor). Similarly, numbers provided in the second paragraph of the results section are not in line with those presented in the first paragraph
Response: Thank you for noticing that, I have reviewed the tables and added the missing numbers.
The paragraphs one says 36 patients underwent surgery in total and the paragraph two says that 35 patients underwent emergency surgery, one had elective surgery and four had their surgery postponed until recovered from COVID-19. I agree that it may be written in a confusing way that is why I have rephrased the sentence.
Reviewer 2 Report
The abstract is not well written
You should include some of the main finding in the abstract section.
Abstract should have a conclusion of the study.
The objective of the study is also not clearly mention.
Add more on the basic of the problem in the introduction
More details about COVID-19 are required in the introduction section.
The author should focus mainly on the importance and significance of the study.
I suggest the author to demonstrate what does the paper add to the current literature? and what new knowledge is added by this study?
Add the unique of this study compared to other studies discuss the same issue.
Discus merits and limitations of technique applied.
The material and method section is too weak in the manuscript and you need to focus on it more.
Author Response
Dear Reviewer,
We appreciate the time and effort that you and the reviewer has dedicated to providing your valuable feedback on my manuscript. We are grateful for your insightful comments on our paper. We have been able to incorporate changes to reflect most of the suggestions provided by you. We have highlighted the changes within the manuscript body (blue font).
Comment: The abstract is not well written
Response:Thank you, it has been revised
“COVID-19 pandemic has made changes to the traditional way of performing surgical consultations. The aim of the present study was to assess the need for surgical care performed by various surgical specialties among COVID-19 infected patients hospitalized in a COVID-19 dedicated hospital. All surgical consultations performed for COVID-19 infected patients in a COVID dedicated hospital in a seven month period have been evaluated. Data on demographics, surgical specialty, consult reason, procedure performed and whether it was a standard face to face or teleconsultation were gathered. Out of 2359 COVID-19 patients admitted to the hospital in the seven month period, 229(9.7%) required surgical care. Out of those 108 consultations which did not lead to surgery 71% have been managed by telemedicine. 36 patients were operated on while suffering from COVID-19. Out of them only 3 patients admitted primarily for COVID-19 pneumonia underwent emergency surgery. Overall mortality amongst operated on was 16,7%. CONCLUSIONS: Patients hospitalised with COVID-19 infection may require surgical care from various surgical specialties, especially during peaks of the pandemic. However, they rarely require a surgical procedure and only occasionally major surgery. Significant portion of potentially surgical problems could be managed by teleconsultations.”
Comment:You should include some of the main finding in the abstract section.
Response:Thank you, it has been added.
Comment:Abstract should have a conclusion of the study.
Response: Thank you, it has been added.
Comment:The objective of the study is also not clearly mention.
Response: Thank you, it has been added.
“The aim of the present study was to assess the need for surgical care performed by various surgical specialties among COVID-19 infected patients hospitalized in a COVID-19 dedicated hospital. As well as assess the possibility of implementing teleconsultations in managing surgical consultations.”
Comment:Add more on the basic of the problem in the introduction
Response:Thank you, it has been revised.
Comment:More details about COVID-19 are required in the introduction section.
Response:Thank you, details about COVID-19 were added to the introduction section
“ COVID-19 is an infectious disease caused by a novel betacoronavirus SARS-CoV-2 [1]. Its clinical manifestations resemble to some extend those of Severe Acute Respiratory Syndrome SARS-CoV-1 and includea range of symptoms like fever, cough, myalgia, diarrhoea or headache [2]. Although the majority of infected patients present mild or no symptoms, a portion progress to respiratory failure with a high fatality rate [2].The novel coronavirus outbreak was first reported in Wuhan, Hubei province, China in December 2019 [3].Since then, it promptly spread to other countries. “
Comment:The author should focus mainly on the importance and significance of the study.
Response: Thank you, it has been revised.
Comment:I suggest the author to demonstrate what does the paper add to the current literature? and what new knowledge is added by this study?
Add the unique of this study compared to other studies discuss the same issue.
Response:It has been added to the discussion section
“To our knowledge this is the first study that explores surgical input of various surgical specialties in the care of patients diagnosed with COVID-19 hospitalised in the COVID dedicated centre. The appropriate management of patients suffering with COVID-19 is still under debate. The pandemic is ongoing with no signs of abating, requiring systemic solutions.”
Comment:Discus merits and limitations of technique applied.
Response: Thank you it has been discussed in the article.
“Thirdly, this is a retrospective study so no interference could be made into the designed clinical pathways. Further, it included different surgical specialties that were at times difficult to compare. “
Comment:The material and method section is too weak in the manuscript and you need to focus on it more.
Response: Thank you, The methodology section has been adjusted.
“Patients flow in the health care system.
During the study period, patients with suspected or confirmed COVID-19 admitted to our centre were either brought in to the emergency department by ambulance or transferred from other hospitals in the Malopolska voivodeship after being diagnosed with COVID-19. The hospital wards were converted into COVID-19 dedicated units with several different profiles. This included dialysis, respiratory, cardiology, surgical wards etc. Patients were assigned to different units depending on their need and comorbidities. The general surgical ward was transformed into a COVID-19 dedicated unit where patients diagnosed with COVID-19 and having concomitant surgical conditions were hospitalised. The patients were managed by the multidisciplinary team comprising of internal medicine doctors and surgical teams of various surgical specialties, depending on the patients need.
Management of surgical consultations
All consultation requests were first reviewed by the consultant surgeons and the decision was made for whether the patient required a face to face consultation or if an over the phone consultation could be performed. If the patient required a face to face consultation, it was performed by a senior clinician to minimise the number of people required to perform the consultation. Before seeing the patient, the surgeon donned using FFP2/3 mask, visors, disposable scrubs, double glows, surgical gown, head cup and theatre clogs. After the consultation, the surgeon doffed and showered in the sluice located outside the COVID ward.
Over the phone consultations were based upon diagnostic imaging, available laboratory results, imaging of the ulcer/wound uploaded into patients records if required and information provided by the leading clinician.”
Reviewer 3 Report
Title: Change to surgical consultation from intervention (as only a small percentage of pts got surgical intervention)
Minor: Change moth to Months in title, consolations to consultation
Methodology: Please discuss in a small paragraph the patient flow in the health care system. Whether patients were admitted to a common COVID ward, principally managed by Internal Medicine. Single department ownership of the patient/multispeciality involvement in management. Presence of surgical team, surgical floor/ ward or the absence of the same (as 33 pts had primary surgical concern).
Please include details of IRB
Discussion: Please discuss what was the pro and cons of having 33 principal surgery patients in a COVID ward not manned by surgical team
I disagree with the statement that majority of the planning were done optimally over the phone. Rather it should be mentioned as a limitation. How did the evaluation of wound, breast ulcer, DVT, trauma was done by telephone without direct patient encounter? Do the authors mean that after initial patient encounter by a surgical team the follow up of these (not sick ones) patients was done telephonically?
Furthermore, neurosurgical,T&Oand maxillofacial consultations could be solely performed via telemedicine
Rephrase: I disagree, how did the clinician establish the extent of these entities without any encounter. Do they Imply they used only imaging to guide treatment. Which is difficult to agree upon, and not patient oriented. Even decreasing the number of clinicians, or number of encounters is a valid strategy in assessing patients.
Please include the limitations of not having details on role of comorbidities, medication interactions, role of mimics, interaction [Refer and cite PMID: 32289168, PMID: 32380803
Author Response
Dear Reviewer,
We appreciate the time and effort that you have dedicated to providing your valuable feedback on our manuscript. We are grateful for your insightful comments on our paper. We have been able to incorporate changes to reflect most of the suggestions provided by you. We have highlighted the changes within the manuscript body (orange font).
Comment:Methodology: Please discuss in a small paragraph the patient flow in the health care system. Whether patients were admitted to a common COVID ward, principally managed by Internal Medicine. Single department ownership of the patient/multispeciality involvement in management. Presence of surgical team, surgical floor/ ward or the absence of the same (as 33 pts had primary surgical concern).
Response: The paragraph has been added in the methodology section
“Patients flow in the health care system.
During the study period, patients with suspected or confirmed COVID-19 admitted to our centre were either brought in to the emergency department by ambulance or transferred from other hospitals in the Malopolska voivodeship after being diagnosed with COVID-19. The hospital wards were converted into COVID-19 dedicated units with several different profiles. This included dialysis, respiratory, cardiology, surgical wards etc. Patients were assigned to different units depending on their need and comorbidities. The general surgical ward was transformed into a COVID-19 dedicated unit where patients diagnosed with COVID-19 and having concomitant surgical conditions were hospitalised. The patients were managed by the multidisciplinary team comprising of internal medicine doctors and surgical teams of various surgical specialties, depending on the patients need.
Management of surgical consultations
All consultation requests were first reviewed by the consultant surgeons and the decision was made for whether the patient required a face to face consultation or if an over the phone consultation could be performed. If the patient required a face to face consultation, it was performed by a senior clinician to minimise the number of people required to perform the consultation. Before seeing the patient, the surgeon donned using FFP2/3 mask, visors, disposable scrubs, double glows, surgical gown, head cup and theatre clogs. After the consultation, the surgeon doffed and showered in the sluice located outside the COVID ward.
Over the phone consultations were based upon diagnostic imaging, available laboratory results, imaging of the ulcer/wound uploaded into patients records if required and information provided by the leading clinician.”
Comment:Please include details of IRB
Response: Thank you, It has been included.
The study was approved by the Bioethics Committee of the Jagiellonian University (1072.6120.228.2020).
Comment:Discussion: Please discuss what was the pro and cons of having 33 principal surgery patients in a COVID ward not manned by surgical team
Response: Thank you, for your comment. Patients who presented with a primary surgical need were admitted to the dedicated surgical COVID ward where they have been managed by surgical team along with medical team. It has been added in the methodology section.
“Patients flow in the health care system.
During the study period, patients with suspected or confirmed COVID-19 admitted to our centre were either brought in to the emergency department by ambulance or transferred from other hospitals in the Malopolska voivodeship after being diagnosed with COVID-19. The hospital wards were converted into COVID-19 dedicated units with several different profiles. This included dialysis, respiratory, cardiology, surgical wards etc. Patients were assigned to different units depending on their need and comorbidities. The general surgical ward was transformed into a COVID-19 dedicated unit where patients diagnosed with COVID-19 and having concomitant surgical conditions were hospitalised. The patients were managed by the multidisciplinary team comprising of internal medicine doctors and surgical teams of various surgical specialties, depending on the patients need. “
Comment:I disagree with the statement that majority of the planning were done optimally over the phone. Rather it should be mentioned as a limitation. How did the evaluation of wound, breast ulcer, DVT, trauma was done by telephone without direct patient encounter? Do the authors mean that after initial patient encounter by a surgical team the follow up of these (not sick ones) patients was done telephonically?
Response: Thank you for your comment. The evaluation of the wound in selected cases could be performed remotely using information provided by the leading clinician and available images. Regarding breast ulcer after the initial image assessment the decision was made to continue diagnostic workup in the out-patient clinic setting after the patient recover from COVID-19. The DVT consultation request sent to vascular surgeons was sent erroneously as angiology doctors are responsible for treatment DVT in our hospital. We have included that in the limitation section. Trauma patients in general were assessed by the appropriate surgical team, however in selected cases where there was a need from another surgical specialty it could be performed over the phone.
One must also keep in mind that all those patients were taken care of by experienced clinicians, within University Hospital wards, located in the same building with immediate access to surgical consultation should anything go not as expected, constantly being monitored by medical personnel. In those circumstances, having a trained proxy at the bedside and patient under professional care sometimes requirements for physical contact for certain type of consultations may be lowered temporarily due to reasons mentioned in the introduction (risk of surgery for patient and risk of infection for personnel). This is actually one of the most important outcomes of our study.
Comment:Furthermore, neurosurgical, T&O and maxillofacial consultations could be solely performed via telemedicine
Rephrase: I disagree, how did the clinician establish the extent of these entities without any encounter. Do they Imply they used only imaging to guide treatment. Which is difficult to agree upon, and not patient oriented. Even decreasing the number of clinicians, or number of encounters is a valid strategy in assessing patients.
Response: It has been added to the discussion section
Comment:Please include the limitations of not having details on role of comorbidities, medication interactions, role of mimics, interaction [Refer and cite PMID: 32289168, PMID: 32380803
Response: This has been included in the limitation section.
“Lastly, the role of comorbidities, medication interactions and the influence of specific medications on clinical outcome of COVID-19 were not investigated in this study. “
“The lower mortality rate in our study could be explained by significant discrepancies in the sample sizes as well as the heterogeneities of the groups. In addition, the role of comorbidities, medication interactions were not included in our study [13,14]. “
Reviewer 4 Report
The manuscript written by Rymarowicz et al. presents data on surgical interventions during the COVID-19 pandemic and describes how surgical specialties can adapt their service to provide adequate patient care during the pandemic. The authors present data collected from the University Hospital Cracow which was converted into a dedicated COVID-19 hospital in spring 2020.
In my opinion, the key message of the manuscript is that teleconsultations can present a viable alternative for live consults during the COVID-19 pandemic. Based on the data presented this seems to be specifically the case for neurosurgical, T&O and maxillofacial consultations.
Please correct the following spelling errors:
Abstract: “consulations“ instead of consultations, weather instead of whether (page 1)
While the conclusions in the manuscript are sound, the study could be improved by including a few aspects:
- The authors should include some incidence rates of SARS-CoV-2 infections in their local community for the study period. This would help readers to understand how similar the situation of the University Hospital Cracow is compared to their own hospital setting.
- Do the authors have any evidence that patient care improved due to teleconsultation? This would greatly increase the impact of the manuscript. For example, do teleconsultations for COVID-19 positive patients decrease the waiting time? Improve patient outcome? Give COVID-19 patients in remote locations access to specialist services? This aspect should at least be discussed in the manuscript.
- The mortality in the described cohort was lower than in the COVIDSurg collaborative. The authors should discuss this aspect.
- The authors should describe how "teleconsultations" are performed. Who decides if a "teleconsulation" is appropriate? Are there any set criteria or specific questions? This could be valuable information for surgeons who are considering to establish a similar service in their respective hospital.
- The authors do not mention the infection control measures taken in their hospital or if there have been any transmissions of SARS-COV-2 to healthcare workers including surgeons during the study period. While one could argue that this aspect is not within the scope of the manuscript, the inclusion of this information would strengthen the study significantly.
Author Response
Dear Reviewer,
We appreciate the time and effort that you have dedicated to providing your valuable feedback on our manuscript. We are grateful for your insightful comments on our paper. We have been able to incorporate changes to reflect most of the suggestions provided by you. We have highlighted the changes within the manuscript body (red font).
The manuscript written by Rymarowicz et al. presents data on surgical interventions during the COVID-19 pandemic and describes how surgical specialties can adapt their service to provide adequate patient care during the pandemic. The authors present data collected from the University Hospital Cracow which was converted into a dedicated COVID-19 hospital in spring 2020.
Comment:In my opinion, the key message of the manuscript is that teleconsultations can present a viable alternative for live consults during the COVID-19 pandemic. Based on the data presented this seems to be specifically the case for neurosurgical, T&O and maxillofacial consultations.
Response: It has been added to the conclusion section
Comment:Please correct the following spelling errors:
Abstract: “consulations“ instead of consultations, weather instead of whether (page 1)
Response: Thank you for noticing that, corrected
Comment:While the conclusions in the manuscript are sound, the study could be improved by including a few aspects:
The authors should include some incidence rates of SARS-CoV-2 infections in their local community for the study period. This would help readers to understand how similar the situation of the University Hospital Cracow is compared to their own hospital setting.
Response:Thank you for your suggestion, we have included it in the introduction paragraph.
“At this time, there were on average 7.2 new cases per 10 000 habitants per day in the Malopolska voivodeship, making it the second largest epicentre in Poland”
Comment:Do the authors have any evidence that patient care improved due to teleconsultation? This would greatly increase the impact of the manuscript. For example, do teleconsultations for COVID-19 positive patients decrease the waiting time? Improve patient outcome? Give COVID-19 patients in remote locations access to specialist services? This aspect should at least be discussed in the manuscript.
Response:It has been added to the discussion section
“Teleconsultations undoubtedly have some advantages, especially in COVID patients. For example, performing the consultation in a timely manner by avoiding spending additional time on donning and doffing, minimising the risk of SARS-CoV-2 infection among hospital staff and using PPE resources wisely. It could be argued that performing teleconsultation instead of face-to-face assessments may compromise patients’ care and appropriate diagnosis. Therefore, the decision to complete a consultation without assessing the patients should be reserved for selected cases only.”
Comment:The mortality in the described cohort was lower than in the COVIDSurg collaborative. The authors should discuss this aspect.
Response:We have added this in the discussion section
“The lower mortality rate in our study could be explained by significant discrepancies in the sample sizes as well as the heterogeneities of the groups. In addition, the role of comorbidities, medication interactions were not included in our study [13,14]. “
Comment:The authors should describe how "teleconsultations" are performed. Who decides if a "teleconsulation" is appropriate? Are there any set criteria or specific questions? This could be valuable information for surgeons who are considering to establish a similar service in their respective hospital.
Response:Thank you, the explanation has been added in the methodology section
“Management of surgical consultations
All consultation requests were first reviewed by the consultant surgeons and the decision was made for whether the patient required a face to face consultation or if an over the phone consultation could be performed. If the patient required a face to face consultation, it was performed by a senior clinician to minimise the number of people required to perform the consultation. Before seeing the patient, the surgeon donned using FFP2/3 mask, visors, disposable scrubs, double glows, surgical gown, head cup and theatre clogs. After the consultation, the surgeon doffed and showered in the sluice located outside the COVID ward.
Over the phone consultations were based upon diagnostic imaging, available laboratory results, imaging of the ulcer/wound uploaded into patients records if required and information provided by the leading clinician.”
Comment:The authors do not mention the infection control measures taken in their hospital or if there have been any transmissions of SARS-COV-2 to healthcare workers including surgeons during the study period. While one could argue that this aspect is not within the scope of the manuscript, the inclusion of this information would strengthen the study significantly.
Response: Thank you, the infection control measures had been added to the methodology section. Unfortunately, although there were cases of contracting the COVID-19 by surgical team, there are no evidenced that it has happened while caring for infected patients.
Round 2
Reviewer 1 Report
Authors have addressed my comments
Reviewer 4 Report
The authors have replied to all comments and have implemented my suggestions.